# Development of a Knee Joint CT-FEM Model in Load Response of the Stance Phase During Walking Using Muscle Exertion, Motion Analysis, and Ground Reaction Force Data

**DOI:** 10.3390/medicina56020056

**Published:** 2020-01-29

**Authors:** Kunihiro Watanabe, Hirotaka Mutsuzaki, Takashi Fukaya, Toshiyuki Aoyama, Syuichi Nakajima, Norio Sekine, Koichi Mori

**Affiliations:** 1Department of Radiology, Saitama Prefecture Saiseikai Kurihashi Hospital, Kuki, Saitama 349-1105, Japan; watanabe@saikuri.org; 2Department of Radiological Sciences, Graduate School of Human Health Sciences, Tokyo Metropolitan University, Arakawa, Tokyo 116-8551, Japan; sekine@tmu.ac.jp; 3Center for Medical Sciences, Ibaraki Prefectural University of Health Sciences, Ami, Ibaraki 300-0394, Japan; mutsuzaki@ipu.ac.jp; 4Department of Orthopaedic Surgery, Ibaraki Prefectural University of Health Sciences Hospital, Ami, Ibaraki 300-0331, Japan; 5Department of Physical Therapy, Faculty of Health Sciences, Tsukuba International University, Tsuchiura, Ibaraki 300-0051, Japan; t-fukaya@tius.ac.jp; 6Department of Physical Therapy, Ibaraki Prefectural University of Health Sciences, Ami, Ibaraki 300-0394, Japan; aoyamato@ipu.ac.jp; 7Department of Radiological Sciences, Ibaraki Prefectural University of Health Sciences, Ami, Ibaraki 300-0394, Japan; nakajimas@ipu.ac.jp

**Keywords:** finite element, knee joint, equivalent stress, joint reaction force, musculoskeletal model

## Abstract

*Background and objectives:* There are no reports on articular stress distribution during walking based on any computed tomography (CT)-finite element model (CT-FEM). This study aimed to develop a calculation model of the load response (LR) phase, the most burdensome phase on the knee, during walking using the finite element method of quantitative CT images. *Materials and Methods:* The right knee of a 43-year-old man who had no history of osteoarthritis or surgeries of the knee was examined. An image of the knee was obtained using CT and the extension position image was converted to the flexion angle image in the LR phase. The bone was composed of heterogeneous materials. The ligaments were made of truss elements; therefore, they do not generate strain during expansion or contraction and do not affect the reaction force or pressure. The construction of the knee joint included material properties of the ligament, cartilage, and meniscus. The extensor and flexor muscles were calculated and set as the muscle exercise tension around the knee joint. Ground reaction force was vertically applied to suppress the rotation of the knee, and the thigh was restrained. *Results:* An FEM was constructed using a motion analyzer, floor reaction force meter, and muscle tractive force calculation. In a normal knee, the equivalent stress and joint contact reaction force in the LR phase were distributed over a wide area on the inner upper surface of the femur and tibia. *Conclusions:* We developed a calculation model in the LR phase of the knee joint during walking using a CT-FEM. Methods to evaluate the heteromorphic risk, mechanisms of transformation, prevention of knee osteoarthritis, and treatment may be developed using this model.

## 1. Introduction

Finite Element Analysis (FEA) is a method to calculate the response of a structure to an external force, and it can predict the strength and mechanical characteristics of the structure [1,2]. The application of this method to the three-dimensional (3D) structures of the human body, obtained by computed tomography (CT), can predict their strength and stress distribution [3]. This technology is called the CT-based finite element method (CT-FEM) and uses quantitative CT images. This technology is being used to further advancements in orthopedics and dentistry [4,5,6,7]. Furthermore, bone strength can be predicted and evaluated using a 3D bone model for each individual using the corresponding images [8,9,10]. CT-FEM studies have been performed for various parts of the spine and joints [11,12,13], and knee joints [14,15,16,17,18,19] being no exception. In the studies of knee joints, dynamic movements were reproduced using virtual joint angles as well as joint and muscle strength values that were different from the actual values of these material properties in general joint structures, and the input of intra-articular properties was also performed [20,21,22,23,24]. However, these CT-FEMs of the knee joint were not used as a musculoskeletal model incorporating the femur and tibia, which would otherwise include bone information, motion analysis, and the ground response of the same person to mimic real walking.

The purpose of this study was to use CT-FEM to develop a computational model for the load response (LR) phase (the most burdensome phase for the knee joint during walking). In addition, bones were not treated as rigid bodies (cortical and trabecular bones), and a computational model was built using the bone density of the subject’s CT data to predict intra-articular stress and tension in the tibia.

## 2. Materials and Methods

### 2.1. Participant

The focus was the right knee of a 43-year-old man who had no history of osteoarthritis or surgeries of the knee. The imaging ranged from the center of the femoral shaft to the ankle joint. B-MAS 200 (Kyoto Science Co., Ltd., Kyoto, Japan) was installed as the reference phantom from the knee joint to the lower end of the lower leg for radiation density calibration and was simultaneously photographed. The tube voltage, tube current, and slice thickness after reconstruction were 120 kV, 200 mA, and 0.625 mm, respectively. A standard bone image reconstruction filter was used. The imaging was performed using the 8-row CT system, ECLOS (Hitachi, Ltd., Tokyo, Japan). This study was approved by the Ibaraki Prefectural University of Health Sciences Ethical Review Board (No. e199, Date of ethical approval: February 27th, 2019), and the informed consent of the participant was obtained. These study was conducted in accordance with the Declaration of Helsinki.

### 2.2. Gait Analysis

#### 2.2.1. Motion Analysis

Gait was analyzed using a motion analysis system and force plates. The subject walked barefoot on a 5-m ground strip at habitual speeds. The motion capture data was obtained at 200 Hz using an eight-camera motion analysis system (Vicon Nexus software; Oxford Metrics Group, Oxford, UK). Two floor-mounted force plates (Kistler Instruments, Winterthur, Switzerland) were used to obtain the ground reaction forces at a rate of 1200 Hz, and the data were synchronized with the motion capture data. According to a lower extremity model of the Plug-In-Gait marker set, which is a widely standardized marker arrangement for three-dimensional motion analysis [25], reflective markers of 9.5 mm in diameter were placed directly over the bilateral anatomical landmarks: the anterior and posterior superior iliac spines, lateral thighs, lateral femoral epicondyles, lateral shanks, lateral malleoli, calcanei, and the tops of the feet at the base of the second metatarsals. The knee joint angles during the stance phase were calculated using the joint coordinate system [26]. According to Perry [27], the stance phase of the gait was sub-divided into the following five phases based on the ground reaction force data: initial contact (IC, 0% of the stance), LR (16% of the stance), mid stance (MS, 50% of the stance), terminal stance (TS, 83% of the stance), and pre-swing (PS, 100% of the stance).

#### 2.2.2. Electromyography (EMG) 

Before electrode attachment, the skin at the electrode site was swabbed with alcohol. The EMG data were recorded from rectus femoris (RF), vastus medialis (VM), vastus lateralis (VL), biceps femoris (BF), and semitendinosus (ST) using Delsys Trigno Wireless Surface EMG System (Delsys Inc, Boston, MA, USA) at a sampling rate of 2000 Hz (bandpass filter: 20 Hz to 450 Hz). EMG activities were full-wave rectified and smoothed using a fourth-order zero-lag low-pass filter (cut-off frequency: 6 Hz). The EMG activity was normalized to the maximum EMG signal obtained during a walking cycle.

### 2.3. Outline of the Model and Calculation Method

The FEM software used for converting CT images of the knee joint into a 3D shape model and joint movement analysis was MECHANICAL FINDER Ver 10.0 (Computational Mechanics Research Center, Tokyo, Japan). The right knee was extracted from the CT images and the extension position image was converted to the flexion angle image in the LR phase. MECHANICAL FINDER can recreate a CT knee image in flexion while preserving the heterogeneous properties of the bone. The quadriceps femoris, biceps femoris, semimembranosus, semitendinosus, and gracilis muscles were set as the muscles that exert tension around the knee joint. The ligaments included the patellar, anterior cruciate, posterior cruciate, medial collateral, and lateral collateral ligaments. The tension exerted by each muscle around the knee joint was calculated using the musculoskeletal modeling system (Any Body Technology, Aalborg). We verified the validity of the simulation using Any Body (Aalborg, Denmark) by comparing the simulated muscle activation with the measured EMG activity.

All calculation methods included static linear analyses that did not consider temporal movements. In this software, the material properties of the joint components were determined using Equations (1)–(5) [28].

#### 2.3.1. CT Value (HU: Hounsfield Unit) Density Conversion Formula

The reference phantom was used simultaneously with the measurement site to calibrate the relationship between the CT value and density. The density of each element (ρ) was calculated from the CT value using the following linear function equation [29,30,31]. The relationship between the CT value and density was approximated by a linear equation, and is expressed by the slope (a) and intercept (b):ρ [g/cm^3^] = (CT value × a) + b,(1)
where a and b were values calculated from the reference phantom. If there is no reference phantom, calibration can be performed using the following equation, which assumes a tube voltage of 125 kV [32]:(2)$$\rho \;[g/{\mathrm{cm}}^{3}]=\{\begin{array}{ll} 0.0 & \left(HU\leq -1\right)\\ \left(\mathrm{CT}\;\mathrm{value}+1.4246\right)\times \frac{0.001}{1.0580} & \left(-1<HU\right) \end{array}.$$

#### 2.3.2. Young’s Modulus

Young’s modulus is a factor that determines the amount of stress required per unit strain within the elastic range of the object, and is also called longitudinal elastic strain. Its value is equal to the stress divided by the strain, as shown in the following formula where E is Young’s modulus [28]:(3)$$E\;\left[\mathrm{MPa}\right]=\frac{\sigma }{\varepsilon }.$$

#### 2.3.3. Young’s Modulus Conversion Formula

In this study, Keyak’s transformation formula was used with reference to previous research [29]:(4)$$E\;\left[MPa\right]=\{\begin{array}{l} \begin{array}{l} 0.001\;\left(\rho =0\right)\\ 33900\rho^{2.20}\;\left(0<\rho \leq 0.27\right) \end{array}\\ \begin{array}{l} 5370+469\;\left(0.27<\rho <0.6\right)\\ 10200\rho^{2.01}\;\left(0.6\leq \rho \right) \end{array} \end{array}.$$

#### 2.3.4. Poisson’s Ratio

In general materials, Poisson’s ratio is in the range of 0.0–0.5; however, in previous studies, Poisson’s ratio of 0.4 has often been used for the bone [29,30]:(5)$$\nu =-\frac{\varepsilon_{2}}{\varepsilon_{1}},$$
where ε_1_ is the transverse elastic strain and ε_2_ is the longitudinal elastic strain.

### 2.4. Modeling Construction and Procedure

CT images were exported into the Digital Imaging and Communications in Medicine format, which were then imported into the software. While setting the range of analysis, the upper limit was the distal end of the femur, approximately 10 cm above the knee joint, and the lower limit was the distal end of the tibia and fibula just above the ankle joint. The range of each data point was visually judged and set from the recreated 3D image.

In the construction of the FEM, a regular tetrahedron element (one side length: approximately 1.0–2.0 mm) was used as a solid element for the bone part. Furthermore, a triangular plate-shaped shell element with 1.0-mm thickness was placed to cover the patellar surface. Previous studies have examined the sizes of the solid and shell elements [31,33]. The sizes recommended by these studies were adopted. Additionally, when applied to the anatomical structure of a bone, the shell element corresponds to the compact bone—the surface part of the bone—and the solid element corresponds to the cancellous bone underneath the surface. The shell element can be placed on the surface of the analysis target to improve the stability of the calculation.

The soft tissue elements, menisci, ligaments, and cartilages were created with the assistance of a 3D modeling software, Metasequoia (Tetraface, Inc., Tokyo, Japan). In addition, the length of one side was about 0.5–2.0 mm, and in solid and shell elements, the ligaments were resistant to bending and compression, so only when stretched using truss elements (line elements) did the model apply. However, since the truss elements concentrate the load at the nodal point, the patellar surface was made into a shell element (thickness: 1.0 mm; Young’s modulus: 1000 MPa) to distribute the large force of the quadriceps femoris on the whole patella to prevent breaking of the patella. The Poisson’s ratio of the shell element was 0.4, and the length of one side was approximately 1.0–2.0 mm. 

#### 2.4.1. Determination of Material Properties

For the material properties of bone elements, Keyak’s conversion formula was used [12,34,35]. In this conversion formula, a linear relationship is assumed between the CT value and bone density, and the bone density is calculated from the CT value. Furthermore, Young’s modulus and yield stress were calculated from the obtained bone density. Here, Poisson’s ratio was taken as 0.4, the material-specific constant value for bone. When calculating the bone density, the phantom for CT value calibration was used to perform calibration using MECHANICAL FINDER (version 10.0).

#### 2.4.2. Determination of Walking Posture

The load on the knee joint during walking has the highest amount of stress in the LR phase. For an accurate model, a motion analysis device was used to analyze the angle between the tibia and floor in the LR phase, the joint angle between the femur and tibia, muscle strength, segment length, and the direction of the external force.

#### 2.4.3. Determination of Load and Restraint Conditions

In our calculation model, we set a load side and a restriction side. A load surface is a surface on which a force acts when a load is applied to a calculation model. Additionally, the direction and magnitude of the force were set by specifying the area to determine the direction, amount, and magnitude of the action. A constraining surface is a surface that supports the load in a part of the calculation model. The calculation model built by MECHANICAL FINDER is created under weightless space conditions. Therefore, when applying a force to the analysis target, a portion (restraint surface) for fixing the target is required. Constraint conditions were such that the proximal end of the femur was completely constrained and the distal ends of the tibia and fibula were constrained in the direction of the X and Y axes. The reaction to the sum of the floor reaction force and each muscle force is the joint reaction force, which is the origin of the stress on the joint. In a model in which either the X- or Y-axis is constrained, stable analysis cannot be performed; therefore, both axes were constrained and calculation was performed using only the vertical component. As a result, in this calculation model, only the Z-axis direction (vertical direction) of the floor reaction force was evaluated [36].

Regarding the traction of the quadriceps femoris muscle, since the quadriceps femoris muscle does not cause an inappropriate deformation due to the tension of the truss element, it is not actually the traction force of the quadriceps femoris but rather forced displacement. The amount of displacement was 3 mm. A strain–tension table—a physical property value of the truss element—was defined so that the tension of the truss element matches the traction force of the quadriceps. Muscle tractive force was calculated using a musculoskeletal modeling system.

The contact conditions were set up for the femur and menisci, femur and tibia, tibia and meniscus, and cartilage between the femur and patella. The coefficient of friction in the contact analysis was 0.01. However, the innermost rim of the meniscus and the tibial cartilage were fixed to account for the protrusion of the meniscus under load.

## 3. Results

Table 1 summarizes the physical properties of the tetrahedral elements used for the anatomical elements. Additionally, Figure 1 illustrates a finite element model of the knee joint. The knee is inverted during the walking stance phase and the lateral joint space is wide open. The angle represents the LR phase and is set based on Table 2, which summarizes the angle between the femur and tibia and the tibia and floor. Figure 2 illustrates forced displacement (white solid arrow) of the quadriceps femoris and the model that set the load and restriction. Table 3 summarizes the material properties of each muscle and ligament that were derived using a truss element. Table 4 summarizes the traction power of muscles (calculation level) and the floor reaction (measured value). Figure 3 illustrates the joint reaction force applied to the femur and tibial surface in the LR. In Figure 3, the C-shaped region over the inner edge corresponds to the medial meniscus. Joint reaction forces are distributed over a wide area of the femur and tibial medial aspect in the LR period to support the load. Furthermore, the resultant joint reaction force was 1981.22 N at the distal end of the femoral cartilage. The direction of the reaction force is outward in the meniscal region and inward in the tibial articular cartilage region. Equivalent stress is illustrated in Figure 4. The range of the stress value was 0–5 MPa. As illustrated in Figure 4, stress was distributed over the entire tibial medial surface and medial meniscus. The equivalent stress distribution area corresponds to the contact reaction force distribution area and is highlighted by the rough surface of the medial upper surface of the femur and tibia and a wide area over the medial meniscus (Figure 4). It shows the total contact area and the average and maximum contact pressure. For reference, the total contact area was 577.718 mm^2^, the average contact pressure 2.50549 N/mm^2^, and the maximum contact pressure 8.95098 N/mm^2^.

Figure 5 shows the simulated muscle activation of the RF (a), VM (b), VL (c), BF (d), and ST (e) muscles and the measured EMG activity of RF (f), VM (g), VL (h), BF (i), and ST (j) muscles during a walking cycle. We confirmed that there was some consistency between the simulated muscle activation and the measured EMG activity patterns.

## 4. Discussion

We constructed a CT-FEM in the LR phase during walking using gait analysis and musculoskeletal modeling with joint components as the material characteristics. Although the results of the CT-FEM analysis of the knee in static posture have been previously reported [37,38,39,40], there are no reports of accurate model methods based on gait data. In this study, the reproduction of the walk is reliable. Using the motion analysis system, the results of the musculoskeletal modeling system and the EMG data approximated, and the results were similar to the EMG patterns obtained in other studies [41]. In terms of kinematics, the LR moves in the flex direction (18°), and the LR moves in the varus direction. In terms of the front view, even in normal cases, varus was observed because of their O-legs Oatis’s book. Further, when the medial pressure is in the one-leg standing state, the load passes through the medial condyle of the knee joint; therefore, the medial pressure increases [42]. By changing the degree of varus, weight, angle, and force of the knee, it may be possible to predict the magnitude and area of stress applied to the joint. 

Emphasis was placed on setting the balance of force to match the analysis results of the musculoskeletal modeling system. To avoid computational errors caused by the rotational movements of the knee, resulting from the moments of muscle traction and the floor reaction force, the analysis was performed by constraining the distal end of the tibia in the X and Y axes. The reason for generating a movement in the FEM may be different from the calculation settings of the musculoskeletal modeling system. The musculoskeletal modeling system is an actual model derived from the movements of the human body, but the FEM does not include the proximal part of the femur and the distal part in the lower femur, and the number of set muscles is calculated by the musculoskeletal modeling system, which may be different from the settings. Alternatively, since the musculoskeletal modeling system is a calculation system that assumes a dynamic state, it may be different from the static (or quasi-static) state that was used in FEM calculations.

In this study, the non-bone tissue was treated as a homogeneous model. However, the meniscus was a mechanical anisotropy model incorporating collagen fibers into the meniscus, which was an issue to consider in calculations. However, we chose the isotropic model. This is because it was not necessary to evaluate the meniscus, and the posture was maintained still and analyzed at a single angle, where the effect of only the vertical component of the three-dimensional coordinate object was observed. Furthermore, FEM calculations have been simplified for clinical use.

The calculation results demonstrate the effects of normal stress. The horizontal component (shear component) is expected to be approximately 20% of the vertical component [43]. Therefore, we observed the main component of the load effect. Considering the contribution of the horizontal component, the equivalent stress will also increase to the same extent. However, it is unclear whether these changes contribute to shear stress. In the future, it is necessary to construct an improved model that can calculate the contribution of the horizontal component. The equivalent stress diagram and the joint reaction force diagram were generated. Equivalent stress refers to the absolute stress value when the multiaxial stress state in 3D space is regarded as a single axis, without distinction between compression and tension. It was used for calculating compression failure and determining the plastic state. Equivalent stress was calculated by the Drucker–Prager yield criterion for bone and the von Mises yield criterion for cartilage. The joint reaction force indicates a reaction force generated on the contact surface between materials by contact analysis. The contact reaction force map is useful in confirming the range of the contact surface of the constructed model.

Equivalent stress was strongly applied to a wide area from the center of the medial condyle in the knee joint. A wider area of equivalent stress was observed in the intercondylar fossa. It is thought that the proximal part of the knee toward the femur was restrained, the internal condyle was compressed, and tensile stress was generated. A point with a considerable amount of stress was observed in the central front part of the tibial side, but this may have occurred because of the slightly convex shape of the bone and the meniscus edge above it. In a normal knee, the arc of the femoral condyle is maintained and, therefore, the force is distributed and received by the menisci and articular cartilages. If the pathological model of osteoarthritis can be constructed using this calculation model and the healthy joint data, a model that can set the weight, club angle, and coefficient of friction between the meniscus and articular cartilage can be constructed. The process of osteoarthritis can be calculated. Using this model, it may be possible to develop assessments of risk, mechanism of transformation, prevention, and treatment of osteoarthritis of the knee.

Based on the abovementioned data, this model is a model that accurately incorporates walking data. Further, in our study, the bone tissue was not a rigid homogeneous model, but a heterogeneous model obtained from the bone density of the subject himself; thus, the model is a more accurate custom-made model. The calculation result is valid because a high-stress concentration occurs inside the articular cartilage and compact bone in a state where the outside of the knee is widened and the load is concentrated inside.

This study had some limitations. Although the model was analyzed and created in detail, only one case was evaluated. In the future, it is necessary to analyze many samples and verify the validity of this method. Further, only the LR phase—the most stressed phase of the knee joint during walking—was evaluated in this study; in the future, the cause, prevention method, and treatment of knee osteoarthritis may be clarified by analyzing those not only in the LR phase but also in other phases. Finally, pre-stress or pre-strain in ligaments is an important factor playing a significant role in the prediction of an accurate ligament force distribution during walking. However, this factor was not incorporated in our knee model. 

## 5. Conclusions

CT-FEM was used to construct a calculation model for the prediction of intra-articular stress and strain in the LR phase, which is the most-stressed phase of the knee joint during walking. The FEM of the knee joint is complicated; however, by combining a motion analysis device, floor reaction force meter, and muscle traction calculation, a more accurate FEM can be constructed.

The contact reaction force and equivalent stress in a healthy individual were determined. The stress is strongly distributed over a large area around the inside of the joint. Comparisons with a similar knee osteoarthritis model can help elucidate the state of progression in osteoarthritis using this high accuracy CT-FEM.

## Figures and Tables

**Figure 1 medicina-56-00056-f001:**
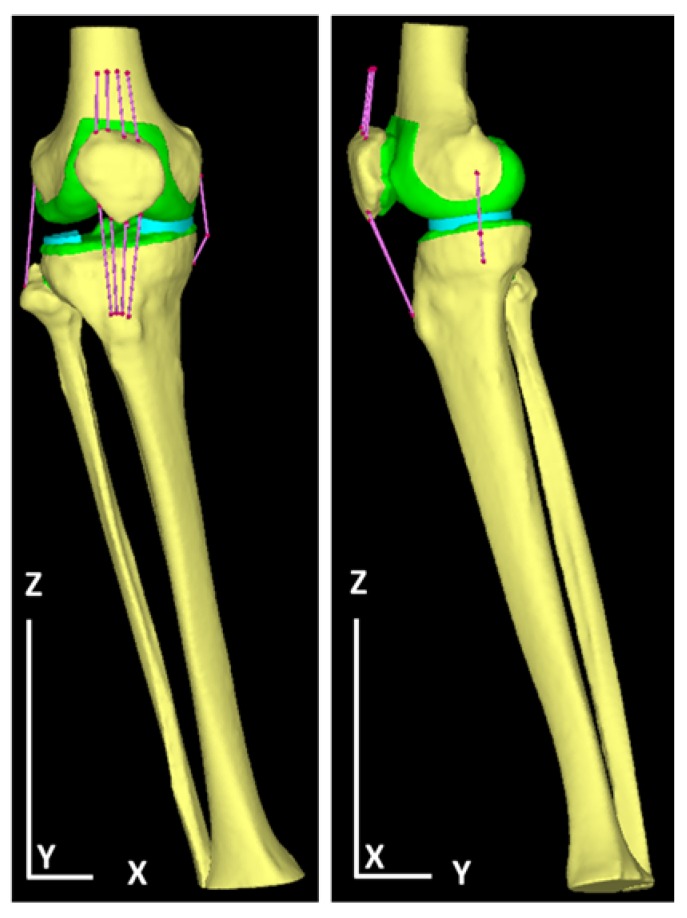
A finite element model of the right knee of a walking posture in the load response (LR) phase in a healthy person. On the left is a front view and on the right is a side view. Bone elements are yellow, truss elements are pink, the meniscus is light blue, and articular cartilage is green. Articular cartilage was set on the femur, tibia, and patella.

**Figure 2 medicina-56-00056-f002:**
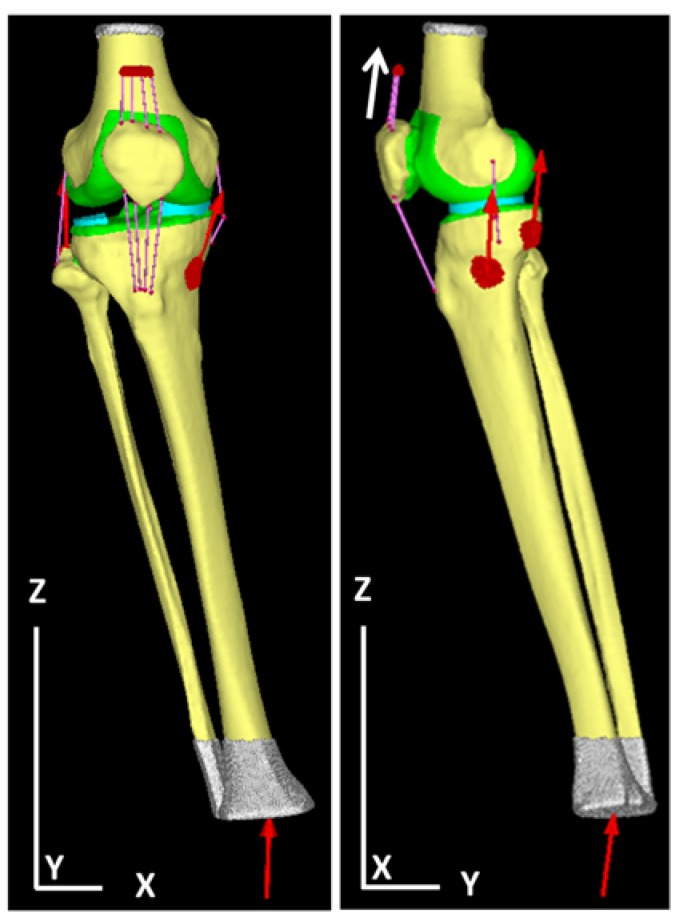
The setting of load and restraint condition in the right knee model. The femoral side was restrained and the quadriceps muscle was forced displacement (↑). The muscle was input in each direction, the lower end of the lower leg was vertical, and the load obtained in the gait analysis was input.

**Figure 3 medicina-56-00056-f003:**
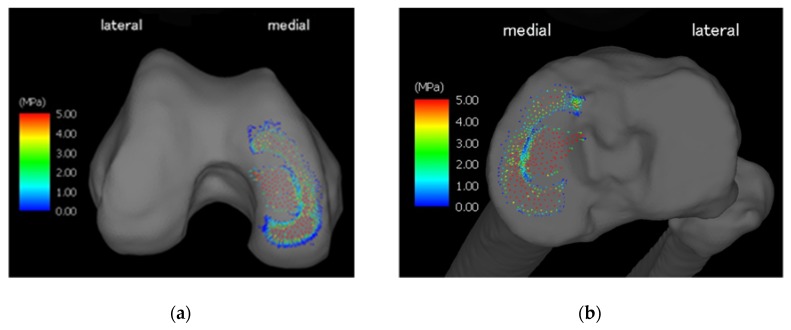
Image (**a**) shows the joint reaction force on the femoral joint surface, and (**b**) shows the joint reaction force on the tibial joint surface. Arrows in the figure indicate the magnitude and direction of the reaction force.

**Figure 4 medicina-56-00056-f004:**
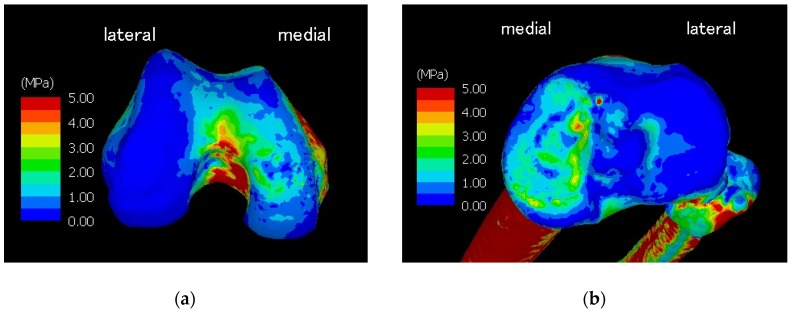
Image (**a**) shows the equivalent stress diagram on the femoral joint surface, and (**b**) shows the equivalent stress diagram on the tibial joint surface. The left side of the figure is the medial side. The range of stress values is 0–5 MPa; it was displayed in blue, green, yellow, and red in the low- to high-stress side.

**Figure 5 medicina-56-00056-f005:**
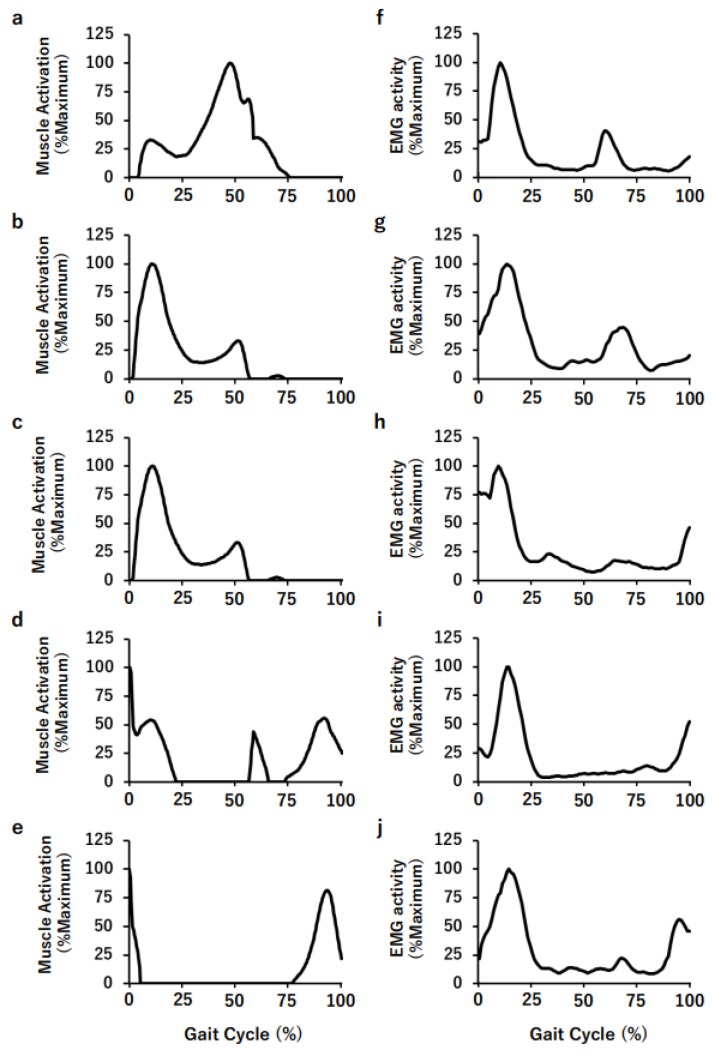
Simulated muscle activation patterns of the RF (**a**), VM (**b**), VL (**c**), BF (**d**), and ST (**e**) muscles, and the measured EMG activity of RF (**f**), VM (**g**), VL (**h**), BF (**i**), and ST (**j**).

**Table 1 medicina-56-00056-t001:** Material properties of the anatomical elements.

Anatomy Element	Young’s Modulus (MPa)	Poisson’s Ratio
Femur, tibia, fibula, patella	Keyak’s conversion formula	0.4
Cartilage	20 (100 only on the fibula)	0.4
Meniscal	20	0.4
Ligament	0.1	0.4

**Table 2 medicina-56-00056-t002:** The angle between the femur and tibia, and the tibia and floor.

**The angle between the leg bone and the floor (deg.)**		
Inversion direction 11.5	Bending direction 13.3	Inward direction 7.7
**The angle between the femur and tibia (deg.)**		
Inversion direction 12.21	Bending direction 14.62	Inward direction −3.28

**Table 3 medicina-56-00056-t003:** Material properties of the muscles and ligament (truss element).

Anatomical Element	Strain Range	Tension (N)
Cruciate ligament, collateral ligament	ε < 0.0 0.0 < ε	0.0 1000ε
Patellar ligament, quadriceps tendon	ε < 0.0 0.0 < ε < 0.005 0.005 < ε	0.0 172754ε 863.77

**Table 4 medicina-56-00056-t004:** The traction force of the muscle (calculated value) and floor reaction force (measured value).

**Floor reaction force (N)**	
Inward direction	26.08
Forward direction	−112.00
Upward direction	808.95
**Muscle traction (N)**	
Quadriceps	863.77
Biceps femoris	266.26
Semimembranosus	99.99
Semitendinosus + gracilis	61.11

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
