# Peer review of "Development of a Knee Joint CT-FEM Model in Load Response of the Stance Phase During Walking Using Muscle Exertion, Motion Analysis, and Ground Reaction Force Data"

_medicina, 2020, doi:10.3390/medicina56020056_

Round 1

Reviewer 1 Report

I have not further comments

Author Response

添付ファイルをご覧ください。

Reviewer 2 Report

The right knee of a 43-year-old man who had no history of osteoarthritis or surgeries of the knee was examined. And a CT-FEM was used to construct a calculation model for the prediction of intra-articular stress and strain in the LR phase. Overall, there are major concerns that cast doubt on the reliability of predictions and biofidelity of the model itself. These issues were unfortunately not presented and critically discussed. As a journal reader, confidence on the relative accuracy of a model is of prime concern. The entire paper needs also to be substantially revised in light of existing shortcomings.

Major comment:

Introduction was somewhat brief and vague and even sometimes wrong. Author's claim that there are no highly-detailed knee CT-FEM model in the literature (lines 58-61) However, FE models with different degree of precision and refinement are already developed and extensively validated (please see : Marouane et al. 2015 JBM; Bendjaballah et al., 1995, The Knee, 2, 69-79). Relatedly, I think it is good practice to not simply refer to a model as “highly-detailed”. It would be more informative to tell readers that the validity of the model has been investigated and compared to other existing models, then present some brief details on why the level of validity is sufficient for the application of the model in this study. Without greater clarity on the Introduction, it was difficult for me to be enthusiastic about the rest of the manuscript. The natural meniscus, similar to the articular cartilage, is a nonhomogeneous tissue reinforced with circumferential and radial collagen fibrils. It also demonstrates distinct response in tension versus compression. Authors should justify the rationale behind the choice of an isotropic elastic material and compared their results with some computational knee joint models that incorporated the collagen fibrils in menisci (Shirazi et al. 2008; Marouane et al. 2015 JBM). Pre-stress or pre-strain in ligaments represent an important factor playing significant role in the prediction of an accurate ligament forces distribution during rotation. Moreover, the absence of this factor can affect the kinematic and the kinetic of the joint during any kind of simulation. Here, I think the authors omitted this factor during their analyses and it will be great if they report this point as limitation to this study. Figure 4: Peak contact stress is not the best available measure to validate a model. Due to the high sensitivity of maximum contact pressure with the mesh density and shape, it is usually recommended to concentrate on mean contact areas and pressures instead. I encourage authors to summarize the extent of validity of the model with available data from computational and experimental studies. This will inform the reader and help establish confidence in model predictions. Does this model valid (for practical purposes) to predict knee kinematic/kinetics response, and/or contact forces, contact areas, etc... Authors did not check if the results are realistic or not, any comparison with experimental data? The Discussion requires further development. Rather than re-stating results, discuss how some of the model assumptions and results relate to our understanding of joint function. Please provide a discussion of how the limitations would have affected the results and ultimate conclusions. For example, how sensitive is the model to different healthy gait paths? What effect does using a gait path from a different subject have on the current model? How are muscle forces calculated? Is it a constrained optimization problem? Does your approach consider the whole lower extremity and simultaneously calculates the muscle forces acting around hip, knee, and ankle? It will be great if the authors provided such EMG data as part of a figure. This will provide the opportunity to the reader to objectively assess muscle force predictions against EMG data, to evaluate the timing as well as relative magnitude. The joint rotation angles are explicitly specified in the finite element analysis. First, were the coordinate systems definitions exactly the same in between the gait analysis and FE model? Such mismatches can significantly effect how motion is described and later applied to different models; see Piazza and Cavanagh (2000). Can the authors provide a sensitivity analysis to establish confidence in model predictions? Bony Structure are much larger stiffer than soft tissue. Hence, Tibia, femur and Patella are generally considered as a rigid body in FE models. How Authors explain the use of a solid elements to represent those structure? To be noted that the tetrahedral elements are stiffer and can give an erroneous result, and there are expensive and time consuming. Furthermore, how authors can explain the Von-Misses stress shown in Figure 4. Normally, during the stance phase of gait the stress should be more pronounced in the cartilage (contact zone of the Tibio-femoral joint) rather than bones. One reason is the constrained tibia in X and Y directions. Can authors discuss on the contribution of passive structure on their results? Figures: I’ll be better to improve figures quality (axes, legends...)

Minor comment:

It has been shown in the literature that treadmill gait can differ from over ground gait in several biomechanical parameters (Murray et al., 1985; Strathy et al., 1983; Warabi et al., 2005; Alton et al.,1 998). In this study, the knee kinematics was studied during treadmill gait, not during over ground walking. Such limits should be mentioned. Moreover, the validation of the musculoskeletal model is a critical point that surely deserves to be discussed more in depth. Line 111: There are no formulae. It is also recommended to use number for each formulae and, hence, referee to it by a number (instead of “following” …). Line 125-128: Please remove or shrink this paragraph, as it contains a general information for readers. Line 136-143: Please remove or shrink this paragraph, as it contains a general information for readers. Line 165-166: Please use “solid elements and “shell elements” instead of “solids” and “shells”. Line 166-167: Is it the number of element in contact? Is it important to see ! How many musculotendon units considered in the model?

Round 2

Reviewer 2 Report

No comment

This manuscript is a resubmission of an earlier submission. The following is a list of the peer review reports and author responses from that submission.

Round 1

Reviewer 1 Report

The authors describe a CT-FEM knee model, while this paper is interesting it lacks an applied sense – aka state clearly what can be done with it. The written language could be improved and the usage of abbreviations could be reduced.

The authors often use the term predict but not prediction or validation was done.

What was the recording frequency of force plate and motion capture – was mocap used to get positions of tibia and femur? What markers were used to determine the tibia and shank position.

What was the self -selected speed

Loading response (LR) seems not defined in the main text body, “MF” is only used 3 times so consider using the full term, “Any Body” could be confused with <AnyBody Technology, Aalborg > consider using “modelling software” and DICOM is only defined but not used.

Line 29: consider using “any computed”

Line 30: please use a “,” rather than “-”

Line 34: use was

Line 35: use were

Line 53: give reference

Line 110: how were a and b calculated?

Line 118: terms in equation are not defined

Line 149: the ligament cannot show resistance it can generate

Line 277: why was this not done?

Line 291: the sentence starting “In the future” is not relevant to the paper

Line 298: use many rather than lots of